# Identification of Vinyl Sulfone Derivatives as EGFR Tyrosine Kinase Inhibitor: In Vitro and In Silico Studies

**DOI:** 10.3390/molecules26082211

**Published:** 2021-04-12

**Authors:** Thitinan Aiebchun, Panupong Mahalapbutr, Atima Auepattanapong, Onnicha Khaikate, Supaphorn Seetaha, Lueacha Tabtimmai, Chutima Kuhakarn, Kiattawee Choowongkomon, Thanyada Rungrotmongkol

**Affiliations:** 1Biocatalyst and Environmental Biotechnology Research Unit, Department of Biochemistry, Faculty of Science, Chulalongkorn University, Bangkok 10330, Thailand; thitinan1906@gmail.com; 2Department of Biochemistry, Faculty of Medicine, Khon Kaen University, Khon Kaen 40002, Thailand; panupma@kku.ac.th; 3Department of Chemistry and Center of Excellence for Innovation in Chemistry (PERCH-CIC), Faculty of Science, Mahidol University, Bangkok 10700, Thailand; iceatima.12@gmail.com (A.A.); onnicha.khai@gmail.com (O.K.); chutima.kon@mahidol.ac.th (C.K.); 4Department of Biochemistry, Faculty of Science, Kasetsart University, Chatuchak, Bangkok 10900, Thailand; supaporn.se@ku.th; 5Department of Biotechnology, Faculty of Applied Science, King Mongkut’s University of Technology of North Bangkok, Bangkok 10800, Thailand; Lueacha.t@sci.kmutnb.ac.th; 6Program in Bioinformatics and Computational Biology, Faculty of Science, Chulalongkorn University, Bangkok 10330, Thailand

**Keywords:** EGFR tyrosine kinase, vinyl sulfone derivatives, in silico study, kinase assay, cytotoxicity assay

## Abstract

Epidermal growth factor receptor (EGFR), overexpressed in many types of cancer, has been proved as a high potential target for targeted cancer therapy due to its role in regulating proliferation and survival of cancer cells. In the present study, a series of designed vinyl sulfone derivatives was screened against EGFR tyrosine kinase (EGFR-TK) using in silico and in vitro studies. The molecular docking results suggested that, among 78 vinyl sulfones, there were eight compounds that could interact well with the EGFR-TK at the ATP-binding site. Afterwards, these screened compounds were tested for the inhibitory activity towards EGFR-TK using ADP-Glo™ kinase assay, and we found that only VF16 compound exhibited promising inhibitory activity against EGFR-TK with the IC_50_ value of 7.85 ± 0.88 nM. In addition, VF16 showed a high cytotoxicity with IC_50_ values of 33.52 ± 2.57, 54.63 ± 0.09, and 30.38 ± 1.37 µM against the A431, A549, and H1975 cancer cell lines, respectively. From 500-ns MD simulation, the structural stability of VF16 in complex with EGFR-TK was quite stable, suggesting that this compound could be a novel small molecule inhibitor targeting EGFR-TK.

## 1. Introduction

Cancer is a devastating disease characterized by uncontrolled growth and spread of abnormal cells and is the second leading cause of mortality worldwide [1]. Nowadays, there are many types of cancer treatment such as chemotherapy, radiation therapy and targeted therapy [2,3,4]. Targeted cancer therapy has become one of the highly effective for cancer treatment due to its specificity towards cancer cells [5]. The overexpression of epidermal growth factor receptor (EGFR) in cancer cells leads to abnormal signal transduction and is closely related to the occurrence of cancer. Therefore, it has become one of the most important protein targets for designing and developing kinase inhibitors that act on oncogenic EGFR [6].

EGFR, a member of the ErbB family of receptor tyrosine kinases, plays important role in cellular signaling pathways, e.g., mitogen-activated protein kinase (MAPK), phosphoinositide 3-kinase (PI3K)/Akt, and signal transducer and activator of transcription (STAT) pathways that regulate key functions such as proliferation, survival, differentiation, and apoptosis [7]. EGFR is composed of an extracellular receptor domain, a single hydrophobic transmembrane region and an intracellular domain, which includes a juxta membrane domain [8], a tyrosine kinase (TK) domain and a C-terminal tyrosine-rich region [9]. The activation of EGFR-mediated signaling pathways begins with EGF binding to the extracellular domain, which activates the TK domain to phosphorylate at its C-terminal tail, and ultimately, initiates downstream signaling pathways [10,11]. Accordingly, targeting EGFR protein has been suggested as a promising strategy for targeted cancer therapy, since the EGFR is commonly overexpressed in many human cancers, including non-small cell lung, head, breast, bladder and ovarian carcinoma [12,13]. Consequently, inhibition of EGFR leads to the inhibition of cancer cells.

The clinically available drugs used as a tyrosine kinase inhibitor of EGFR (EGFR-TKI) such as erlotinib [14] and gefitinib [15]. The erlotinib is widely used in cancer patients for its inhibitory activity against EGFR exon 19 deletions or the L858R mutation [16,17]. However, these drugs have several side effects such as anemia, balance impairment, dizziness and headache. In addition, acquired drug resistance caused by the secondary mutation T790M of EGFR-TK domain develops inevitably after a median response duration of 9 to 13 months [18]. Therefore, the searching for promising compounds effectively targeting mutated EGFR-TK has become an imperative necessity [19,20].

Vinyl sulfone (VF) is an organic compound, where its core structure is similar to that of chalcones [21,22,23,24] (Figure 1). Previous study has shown that chalcone derivatives can inhibit EGFR activity with the IC_50_ value ranged from 10.3–15.4 µM [25]. Thus, we hypothesized that VF derivatives can inhibit EGFR-TK activity in a manner similar to chalcones. In this study, we aimed to find new potential anti-cancer agents against EGFR-TK. A series of designed VF derivatives was initially screened by molecular docking technique. Subsequently, the kinase inhibition assay of the screened compounds against EGFR-TK was studied. Then, the in vitro cytotoxicity assay towards EGFR expressing lung carcinoma cell lines (A549 and A431) and T790M expressing lung cancer cell line (H1975) was conducted using MTT assay. Finally, the molecular dynamics simulation and free energy calculation were performed to investigate the structural and dynamics properties as well as the binding efficiency of the most potent VF in complex with EGFR-TK.

## 2. Results and Discussion

### 2.1. Molecular Docking

Initially, the 78 VFs (Figure 1) were investigated their binding mechanism using the CDOCKER module of Accelry Discovery Studio 3.0. Each compound was separately docked into the ATP-binding pocket of EGFR-TK complex. The interaction energy of erlotinib is −45.49 kcal/mol, while the interaction energy of all studied compounds is ranged from −43.47 to −21.61 kcal/mol. These results suggested that none of the vinyl sulfone derivatives is stronger than erlotinib. So, we cut off the compounds using the interaction energy lower than −37.5 kcal/mol, and we found eight VFs (VF15, VF16, VF29, VF37, VF41, VF52, VF69 and VF71) that showed lower interaction energies than the others (Figure 2). These compounds can interact with important surrounding residues in ATP-binding pocket of EGFR-TK via H-bonding, pi interactions and van der Waals (vdW) forces (Figure 3). Interestingly, the sulfonate group of most compounds exhibited H-bond formation at the hinge region residue M769 (Figure 3A–I) [26,27] similar to erlotinib. The VF29, VF37, and VF41, which have nitro group as substituents, formed H-bond with T830 residue at A loop. Moreover, the matched vdW contacts between all VFs and erlotinib were as follows: (i) hinge region: T766, L768, P770, and G772 and (ii) A loop: K721, E731, T830, and D831. These results suggested that these eight VFs might be the potent candidate compounds acting against EGFR-TK.

### 2.2. Drug-Likeness Prediction

Physical properties of the eight potent VFs were investigated in term of the drug-likeness by considering their physicochemical properties, including molecular weight (MW), the numbers of hydrogen bond donors (HBD) and acceptors (HBA), rotatable bond (RB), polar surface area (PSA) and lipophilicity (LogP) using the SwissADME web server [26]. The obtained results (Table 1) revealed that all VFs showed the acceptable value following the criteria: (i) Mw ≤ 500 Da, (ii) HBD ≤ 5 and HBA ≤ 10, (iii) RB ≤ 10, (iv) PSA ≤ 140 Å, and (v) LogP ≤ 5 [27]. Therefore, these VFs could likely be developed as promising novel EGFR-TK inhibitors.

### 2.3. Inhibition of the EGFR-TK by Vinyl Sulfone Derivatives

Since the data shown above suggested that the eight VFs may be effective against EGFR-TK, we then investigated the EGFR-TKI inhibitory activity of the eight potent VFs and erlotinib at 1 µM using ADP-Glo kinase assay (Figure 4A). Interestingly, we found that VF16 showed the highest EGFR-TK inhibitory activity (98.91%), which was higher than erlotinib (87.80%). Then, VF16 was selected to evaluate the half-maximal inhibitory concentration (IC_50_) values. As shown in Figure 4B, the IC_50_ against EGFR-TK of VF16 is 7.85 ± 0.88 nM, which is significantly lower than the erlotinib (IC_50_ of 26.09 ± 5.42 nM). Additionally, the inhibitory activity of VF16 is greater than that of the chalcones that have been previously reported to inhibit EGFR activity (IC_50_ ranked from 10.3 to 15.4 µM) [25].

In order to confirm the inhibition selectivity of VF16 towards EGFR-TK, we further performed kinase inhibition assay of VF16 against JAK3 and HER2 protein kinases because both kinases are one of the most studied kinase families [28,29]. The obtained results showed that VF16 showed very low inhibitory activity against JAK3 (IC_50_ of 158.45 ± 4.75 nM) and HER2 (IC_50_ of 312.00 ± 0.28 nM) (Appendix A) as compared to EGFR-TK (IC_50_ of 7.85 ± 0.88 nM), suggesting that VF16 was specific to EGFR-TK rather than JAK3 and HER2.

### 2.4. Cytotoxicity

The VF16 was selected to evaluate IC_50_ values against A549 and A431 cell lines overexpressing wild-type EGFR (A549 and A431) and mutant EGFR human lung cancer cell line (H1975) using MTT assay. The obtained results (Figure 5) revealed that the cytotoxic activity of VF16 against A549 (IC_50_ of 54.63 ± 0.09 µM) and A431 (IC_50_ of 33.52 ± 2.57) was similar to that of erlotinib (IC_50_ of 48.21 ± 7.43 µM and 27.19 ± 6.93 µM for A549 and A431, respectively). Note that VF16 inhibited the A431 cells better than A549 cells because (i) the EGFR expression level found in A431 cells is dramatically higher than that found in A549 [30] and (ii) A549 cells exhibits KRAS mutation, which constitutively activates downstream MAPK signaling pathways, causing a compensatory mechanism [31]. In H1975 cells, the VF16 (IC_50_ of 30.38 ± 1.37 µM) was more susceptible than the erlotinib (IC_50_ of 98.93 ± 1.74 µM) by ~3 times. The lower susceptibility of erlotinib towards T790M EGFR-expressing cells (H1975) compared to wild-type EGFR-expressing cells (A549 and A431) is in good agreement with the previous studies [32,33,34]. Altogether, these findings suggested that VF16 exhibited potent anti-lung cancer activity in all three lung cancer cell lines, which could be developed as a novel anti-lung cancer agent.

### 2.5. Molecular Dynamics Simulation

The structural stability of VF16 bound to the EGFR-TK domain was characterized using RMSD calculation plotted along the simulation time, and the obtained results are illustrated in Figure 6A. The RMSD values of VF16 were slightly fluctuated at the first 150 ns and then reached the equilibrium state after 250 ns with an average RMSD value of ~0.5–2 Å. In the case of whole protein and backbone of protein, the RMSD values were slightly fluctuated at the first 100 ns then showed the stable values along the last stage of MD simulation with an average RMSD value of ~2–3.5 Å.

In addition, the MD trajectories of this system were selected for further analysis in terms of: (i) the number of H-bond and (ii) number of contact atom within the 3.5 Å sphere of VF16, respectively. The number of intermolecular hydrogen bonds and intermolecular contacts between VF16 and its surrounding residues was computed along 500 ns MD simulation represented in Figure 6A. According to the results, we found that VF16 formed two H-bonds with the M769 and C797, in which the C797 could form stronger H-bond than the M769. In addition, we found ~10 intermolecular contacts steadily formed between VF16 and EGFR-TK. These findings suggested that our simulation model was stable. In this work, the MD trajectories from 300 to 500 ns were thus extracted for further analysis in terms of Δ*G*_bind_ values (kcal/mol) and key binding residues of VF16 against EGFR-TK.

### 2.6. Binding Affinity and Key Residues for VF16/EGFR-TK Complex

The free energy calculation based on MM-GBSA method was applied to predict the binding affinity of VF16/EGFR-TK complex. We found that the Δ*G*_bind_ of VF16 is almost identical to the Δ*G*_Exp_ (Table 2). To investigate the key residues of EGFR-TK for VF16 binding, the per-residue decomposition free energy (Δ*G*_residue_) based on the MM-GBSA method was applied on the 100 snapshots over the last 200 ns MD simulation. Note that among residues 695–1018, only residues 695–900 are shown in Figure 6B, where the binding orientation of VF16 inside the ATP-binding pocket of EGFR-TK is shown in Figure 6B. The obtained results revealed that there were eight residues (L718, V726, G796, C797, D800, R841, L844, and D855) that were important for the binding of VF16. The binding residues of VF16 observed in this work were also found as a major interaction in erlotinib/EGFR-TK complex (including L718, A743, L792, M793, G796, and L844) [35,36]. Figure 6B showed binding orientation inside the ATP-binding pocket of VF16/EGFR-TK complex, and we found that the -OCH_3_ moiety of VF16 strongly formed H-bond with C797 and weakly formed H-bond with M769. This is in good agreement with the previous reports showing that H-bond formation with M769 is the main interaction of erlotinib and gefitinib in complex with wild-type EGFR and mutant EGFR [37,38,39,40].

## 3. Materials and Methods

### 3.1. Interaction Energies between Vinyl Sulfone Derivatives and the ATP-Binding Site of EGFR-TK by Molecular Docking Technique

The crystal structure of EGFR complexed with erlotinib (PDB ID: 1M17) [39] was downloaded from Protein Data Bank (PDB). The 3D structure of the drug (erlotinib) was obtained from the ZINC database, whilst the 3D structures of vinyl sulfone derivatives were generated using the Gaussian 09 program. Note that the vinyl sulfone derivatives were constructed according to their availability from previous study [21,22,23,24]. All the ligands were optimized using the Gaussian 09 program (HF/6–31d) as per the standard protocol [42,43,44]. The protonation state of all studied ligands was characterized using the ChemAxon [45].

For system validation, the crystalized ligands were defined as a center in the active site for redocking using CDOCKER programs and the results are shown in Appendix A. The docking protocols of EGFR system was set as 15 Å for sphere docking and docked into the binding pocket with 100 independent runs. The binding between protein and compounds/drug was visualized using the Accelrys Discovery Studio 3.0 (Accelrys Inc., Cambridge, UK) and UCSF Chimera package [46].

### 3.2. Predicted Physicochemical Properties

Physicochemical features such as hydrogen bond donors, hydrogen bond acceptors and drug-likeness play an important role in drug discovery and development [47]. Herein, such properties of the potent compounds were calculated in comparison with known drugs (erlotinib) using web-based applications SwissADME (www.swissadme.ch/) (accessed on 19 June 2020) [26].

### 3.3. Chemical Reagents and Cell Lines

The ADP-Glo^TM^ Kinase Assay kit was purchased from Promega (Madison, WI, USA). EGFR and HER2 was obtained from the previous report [48]. JAK3 (SRP0173) were purchased from Sigma-Aldrich (Darmstadt, Germany). The series of vinyl sulfone derivatives were kindly provided by Dr. Chutima Kuhakarn from Department of Chemistry and Center of Excellence for Innovation in Chemistry (PERCH-CIC), Faculty of Science, Mahidol University [21,22,23,24]. Note that, due to limited amounts of vinyl sulfones obtained from previous study, we performed EGFR kinase and cytotoxicity assays as well as binding pattern study at the molecular level on only 78 vinyl sulfone derivatives (Figure 1). The lung carcinoma A549 (ATCC CCL-185) and A431 (ATCC CRL-1555) cell lines were purchased from the American Type Cell Culture Collection (ATCC, Manassas, VA, USA). The EGFR mutated human lung cancer cell line (H1975) was provided by Dr. Chanida Vinayanuwattikun from Department of Medicine, Chulalongkorn University. Dulbecco’s modified Eagle’s medium (DMEM), RPMI-1640 medium, fetal bovine serum (FBS), penicillin-streptomycin (Pen-Strep) and trypsin were purchased from Life Technologies (California, USA). 3-(4,5-dimethylthiazol-2-yl)-2,5-diphenyltetrazolium bromide (MTT) and dimethyl sulfoxide (DMSO) were purchased from Sigma-Aldrich (Darmstadt, Germany).

### 3.4. Inhibition of the EGFR-TK by Vinyl Sulfone Derivatives

The selected sulfone derivatives that had the interaction energy lower or equal than erlotinib and physicochemical properties showed the acceptable value, were screened for their ability to inhibit the tyrosine kinase activity of the EGFR using the ADP-Glo^TM^ kinase assay as previously reported [25,49]. The first 8 µL of buffer (40 mM Tris-HCl pH 7.5, 20 mM MgCl_2_, and 0.1 mg/mL bovine serum albumin) was added to a 384-well plate. Then, 5 µL of EGFR enzymes (1.25 ng/μL) and 2 µL of inhibitors were added, followed by 10 µL of a mixture of 5 µM ATP and 2.5 µM poly(glu-tyr), and incubated for 1 h at room temperature. Next, 5 µL of the ADP-Glo reagent was added and incubated for 40 min, after that, 10 µL of kinase detection reagent was added and incubated at room temperature for 30 min to convert the ADP to ATP. The ATP was then detected by measuring the luminescence using a microplate reader (Infinite M200 microplate reader, Tecan, Männedorf, Switzerland). All assays were performed in triplicate. The relative inhibition (%) of inhibitors were then calculated compared to the control with no inhibitor as shown in Equation (1):(1)$$\%\mathrm{Relative}\;\mathrm{inhibition}=\frac{\left[\left(\mathrm{positive}-\mathrm{negative}\right)-\left(\mathrm{sample}-\mathrm{negative}\right)\right]}{\left(\mathrm{positive}-\mathrm{negative}\right)}\;\times \;100\;$$

From this equation, the positive is the addition of the enzyme in the reaction, while the negative is without the enzyme in the reaction.

### 3.5. Cell Cultures

The A549 and A431 cells were grown in complete DMEM medium supplemented with 10% (*v*/*v*) FBS, 100 U/mL penicillin and 100 µg/mL streptomycin. All cells were maintained at 37 °C in a 5% (*v*/*v*) CO_2_, 95% (*v*/*v*) air humidified incubator while H1975 cells was grown in complete RPMI-1640 medium at 37 °C in a 5% (*v*/*v*) CO_2_, 95% (*v*/*v*) air humidified incubator.

### 3.6. Cytotoxicity in Cancer Cell Lines

The in vitro cytotoxicity activity of vinyl sulfone derivatives against the A549, A431 and H1975 cell lines were evaluated using the MTT assay. The first 100 µL of A549 (5000 cells/well), A431 (5000 cells/well) and H1975 (5000 cells/well) cells suspension was seeded per well in a 96-well microplate and incubated at 37 °C overnight, cells were treated with compounds and known drug (erlotinib) different concentration. Then, incubated for 72 h. Subsequently, the MTT solution (5 mg/mL) was added in A549, A431 and H1975 cells and incubated at 37 °C for 3 h. The medium was removed and 50 µL of DMSO was added to each well to lyse the cells. Finally, the absorbance was measured at 570 nm using a microplate reader (Infinite M200 microplate reader, Tecan, Männedorf, Switzerland).

### 3.7. Molecular Dynamics Simulation

The starting crystal structure of EGFR-TK (PDB ID: 1M17) [39] was obtained from Protein Data Bank (PDB). The 3D structure of vinyl sulfone derivatives and known drug inhibitor (EGFR-TK) of EGFR-TK were generated and optimized HF/6–31G(d) method implemented in the Gaussian09 software [42,43,44]. The protein-ligand complexes were generated using the CDOCKER module accordance the standard protocol [50]. The docked VFs/EGFR-TK complex with lowest interaction energy and binding pattern similar to the erlotinib (Appendix A) was selected as the initial structure for performing the Molecular Dynamics Simulation studies. The electrostatic potential (ESP) charges were consequently calculated with the same level of theory and were then fitted into restrained ESP (RESP) charges using the ANTECHAMBER module of AMBER16 [44,51,52]. The FF14SB [53] and GAFF [42,54] force fields were applied for protein and VF16, respectively. All missing hydrogen atoms of protein and ligand were added using LEaP module and were then minimized in order to remove the bad contacts. Each system was neutralized by the counter ions and immersed in a TIP3P water [55] box that extended at least 13 Å from the protein surface. Afterward, the complexes were energy-minimized by 1500 interactions of steepest descent (SD) and conjugated gradient (CG) methods using AMBER16 with the AMBER ff14SB force field.

The simulations are carried out under periodic boundary condition with NPT ensemble using a time step of 2 fs. The short-range cutoff for nonbonded interactions is set as 10 Å, whilst the Particle Mesh Ewald (PME) summation approach is applied to treat long-range electrostatic interaction [56,57]. Temperature and pressure are controlled by Berendsen weak coupling algorithm. The SHAKE algorithm is used to constrain all covalent bonds involving hydrogen atoms [58]. The simulated models are then heated up to 310 K for 100 ps and are continuously held at this temperature for another 100 ns or until the simulations have reached equilibrium. Finally, the structural and dynamics behaviors of each complex will be analyzed, including root mean square deviation (RMSD), number of H-bonds between the ligand and EGFR-TK and number of contact atom via the cpptraj module [59]. Besides, the MM-GBSA and ΔGbind, residue were calculated by the MM-PBSA.py module [58,59].

### 3.8. Statistical Analysis

The data are represented as mean ± standard error of mean (SEM). Differences between groups were compared using one-way ANOVA, followed by Tukey’s test for multiple comparisons. The differences in means were determined at the confidence level *p* ≤ 0.05.

## 4. Conclusions

This work combined the computational and experiment techniques to identify new EGFR inhibitor based on vinyl sulfone derivatives. Eight vinyl sulfones from molecular docking technique were tested the inhibitory activity against EGFR-TK and cell-based assay in three cancer cell lines (A549, A431, and H1975 cell lines). The results showed that VF16 can inhibit EGFR-TK activity better than the approved drug erlotinib and showed higher cytotoxicity against A431 cell line than A549 cell line. Additionally, showed high cytotoxicity against H1975 cell lines. From MD simulations, our simulation model of VF16/EGFR are stable. In addition, VF16 showed strongest H-bond with C797, and the key residues responsible for VF16 binding were L718, V726, G796, C797, D800, R841, L844, and D855, in which these binding residues were also found in a major interaction between erlotinib and EGFR-TK. Thus, VF16 could be developed as a promising new anti-cancer drug targeting EGFR-TK.

## Figures and Tables

**Figure 1 molecules-26-02211-f001:**
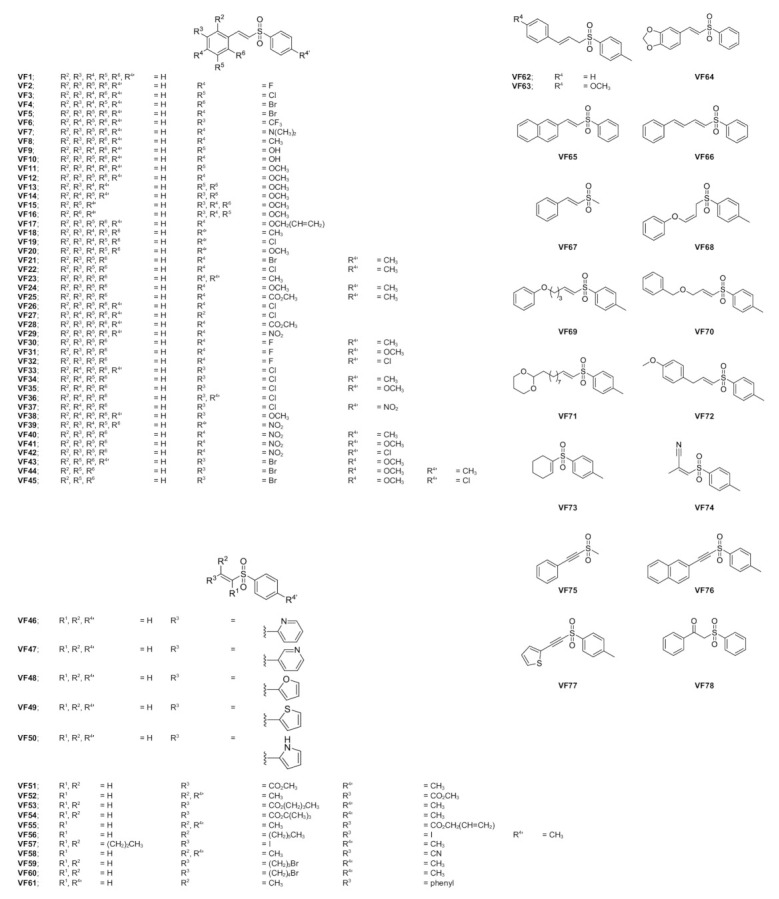
The chemical structures of VFs obtained from the previous study [21,22,23,24].

**Figure 2 molecules-26-02211-f002:**
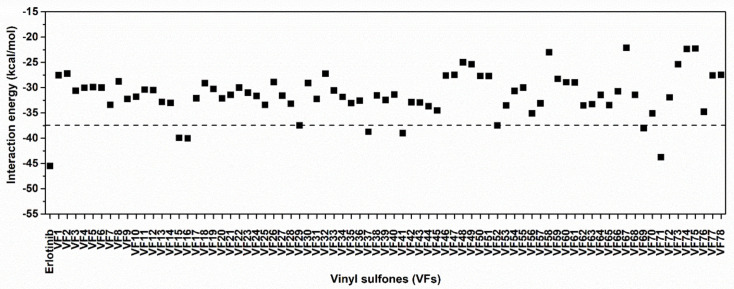
CDOCKER interaction energies of VFs and known drug erlotinib against EGFR-TK at ATP-binding site.

**Figure 3 molecules-26-02211-f003:**
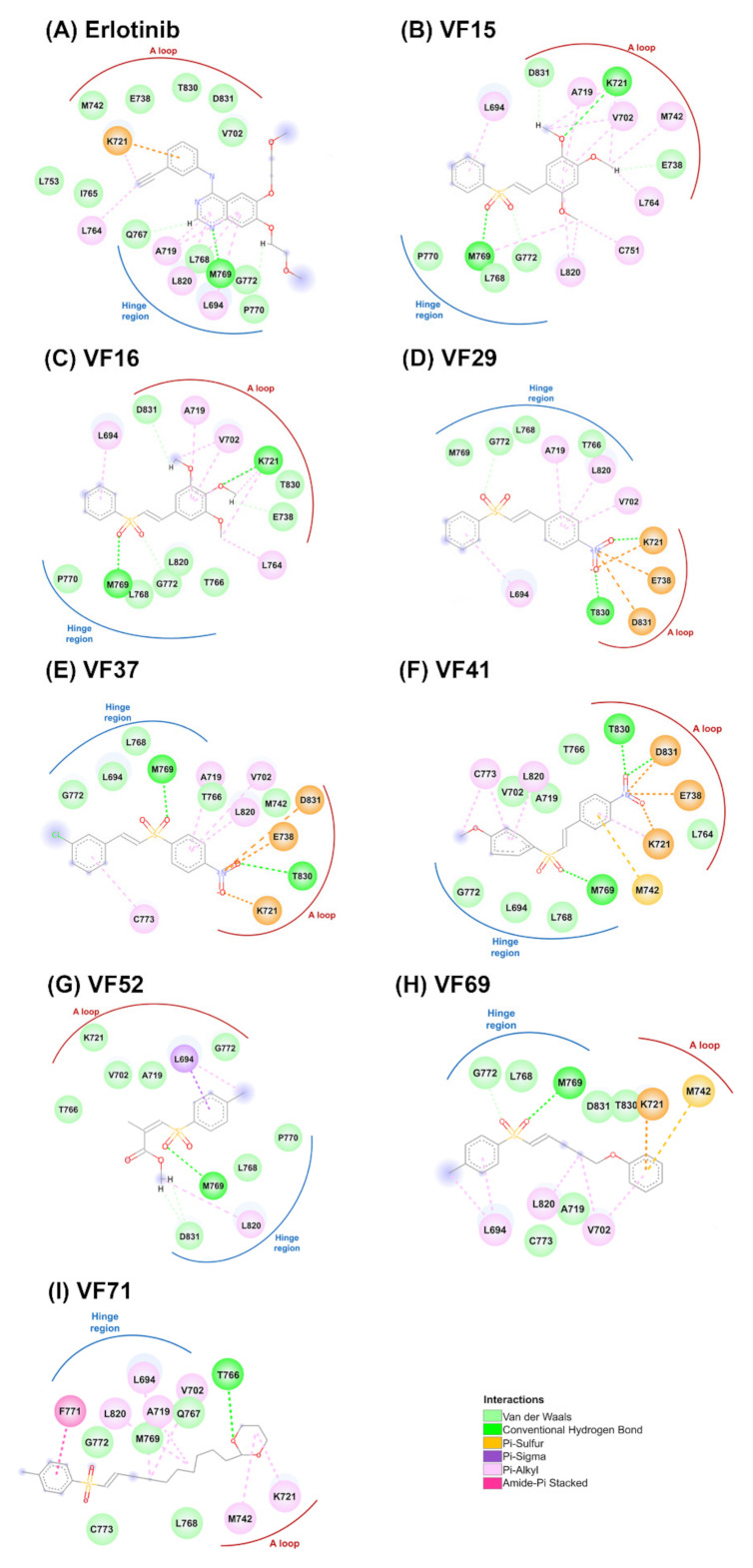
2D interactions of EGFR-TK in complex with erlotinib (**A**) and VFs (**B**–**I**).

**Figure 4 molecules-26-02211-f004:**
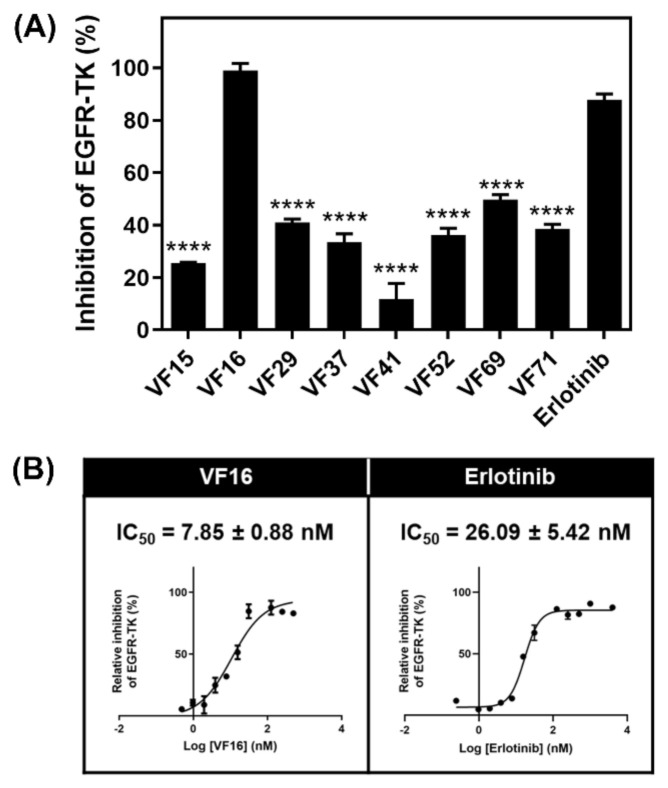
(**A**) Kinase inhibitory activity screening of VFs towards EGFR-TK at 1 μM. **** *p* ≤ 0.0001 vs. erlotinib. (**B**) Kinase inhibitory activity of VFs towards EGFR-TK. Data are represented as means ± SEM from triplicate independent experiments.

**Figure 5 molecules-26-02211-f005:**
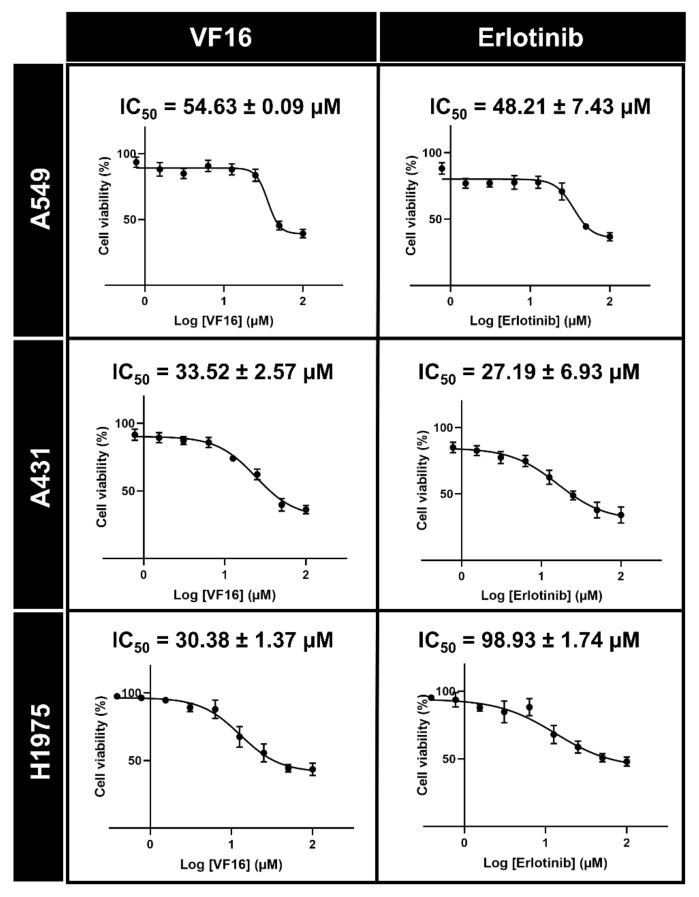
Cytotoxicity in three cancer cell lines (A549, A431, and H1975) after treated with various concentrations of VF16 compared to the known drug erlotinib for 72 h. Data are represented as means ± SEM from triplicate independent experiments.

**Figure 6 molecules-26-02211-f006:**
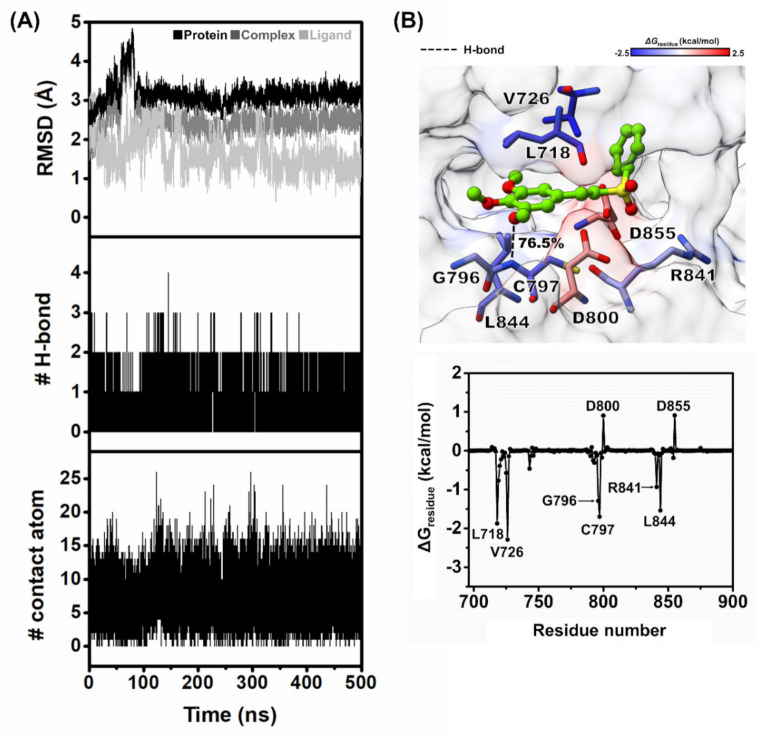
(**A**) All-atom RMSD, number of H-bonds, and number of contacts atom of VF16 in complex with EGFR-TK plotted along the 500 ns MD simulation. (**B**) Binding orientation of VF16 inside the ATP-binding pocket of EGFR-TK. The contributing residues involved in ligand binding are colored according to their Δ*G*_residue_ values, where the highest to lowest free energies are shaded from blue to red, respectively (H-bond interaction represented by black dash line) and per-residue decomposition free energy of the VF16 complex with EGFR-TK.

**Table 1 molecules-26-02211-t001:** Predicted Lipinski’s rule of five for the vinyl sulfones and the known drug. HBD, hydrogen bond donor; HBA, hydrogen bond accepter; PSA, polar surface area.

Compound	Lipinski’s Rule of Five						Drug-Likeness
	MW (≤500 Da)	HBD (≤5)	HBA (≤10)	RB (≤10)	PSA (≤140 Å)	LogP (≤5)	
**Erlotinib**	393.44	1	6	10	74.73	3.20	Yes
VF15	334.39	0	5	6	70.21	2.76	Yes
VF16	334.39	0	5	6	70.21	3.00	Yes
VF29	291.32	2	4	4	86.22	2.32	Yes
VF37	325.77	2	4	4	86.22	2.32	Yes
VF41	321.35	2	5	5	95.45	2.66	Yes
VF52	254.30	0	4	4	68.82	2.38	Yes
VF69	316.41	0	3	7	51.75	3.26	Yes
VF71	380.54	0	4	10	60.98	4.34	Yes

**Table 2 molecules-26-02211-t002:** The MM-GBSA Δ*G*_bind_ and its energy components (kcal/mol).

Energy Component	Δ*G*_bind_ (kcal/mol)
Δ*E*_vdW_	−38.99 ± 0.29
Δ*E*_ele_	−12.95 ± 0.55
Δ*G*_gas_	−51.94 ± 0.68
Δ*G*_solv_	24.69 ± 0.53
Δ*G*_bind_	−8.74 ± 0.32
Δ*G*_Exp_ ^a^	−10.13 ± 0.88

^a^ Experimental binding free energies (Δ*G*_Exp_) was converted from the IC_50_ value using the Cheng-Prusoff equation of Δ*G*_Exp_ = *RT*In(IC_50_) [41].

## Data Availability

Data contained in the manuscript are available from the authors.

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
