# Peer review of "Identification of Vinyl Sulfone Derivatives as EGFR Tyrosine Kinase Inhibitor: In Vitro and In Silico Studies"

_molecules, 2021, doi:10.3390/molecules26082211_

Round 1

Reviewer 1 Report

The manuscript entitled Identification of vinyl sulfone derivatives as EGFR tyrosine kinase inhibitor: In vitro and in silico studies by Aiebchun, T. et al. describes validation of a series of chalcone vinyl sulfone derivatives as potential EGFR tyrosine kinase inhibitors. Molecular docking of 78 vinyl sulfone derivatives identified eight potential ATP-competitive EGFR kinase inhibitors. These eight compounds where tested in biochemical and whole cell assays, and one potent inhibitor (VF16) was identified. The VF16 was further tested in a cytotoxicity assay. Furthermore, 500-ns MD simulation of VF16 in complex with EGFR was carried out.

Further experiments are required to support the claims made by the authors and fundamental information about the compounds are missing. The manuscript will be of some interest to the readers of Molecules.

The manuscript qualifies for publication in Molecules after major revision.

Line 68-74: there is no information about how the different compounds used in this study were selected. What parameters were used?

Line 81-83: the figure does not add any useful information to the manuscript.

Line 84: the molecular docking data need to be discussed in much more detail- what do the best eight compounds have in common? How do they interact with the ATP-binding site? An illustration would be appropriate.  

Line 102-104: It would be much more informative to see how the compounds dock to the protein structure than a listing of the contact residues.

Line 128: figure 4A shows single concentration inhibition at 1µM and erlotinib gave 57% inhibition. However, the IC50 for erlotinib was determined to 26 nM (figure 4b). It is surprising that the 1 µM concentration only lead to 57% inhibition (38-fold higher concentration than the measured IC50 value).

Line 134-137: the authors report IC50 values as a measure for cytotoxicity-this is not correct (IC50 = half-maximal inhibitory concentration). In section 3.6. Cytotoxicity in cancer cell lines, the use of an MTT assay is described. However, the data from this assay is not shown or discussed in the manuscript.

The manuscript does not include any selectivity date for VF16. Compound selectivity is an important issue when developing new bioactive compounds. In many instances, a lack of selectivity can translate to increased toxicity. Protein kinases are particularly concerned with this issue because they share high sequence and structural similarity. The selectivity of VF16 should be assessed by using a protein kinase-profiling panel.

There are several typographical errors in the manuscript. The authors should find a native English speaker to proofread the manuscript.

The manuscript entitled Identification of vinyl sulfone derivatives as EGFR tyrosine kinase inhibitor: In vitro and in silico studies by Aiebchun, T. et al. describes validation of a series of chalcone vinyl sulfone derivatives as potential EGFR tyrosine kinase inhibitors. Molecular docking of 78 vinyl sulfone derivatives identified eight potential ATP-competitive EGFR kinase inhibitors. These eight compounds where tested in biochemical and whole cell assays, and one potent inhibitor (VF16) was identified. The VF16 was further tested in a cytotoxicity assay. Furthermore, 500-ns MD simulation of VF16 in complex with EGFR was carried out.

Further experiments are required to support the claims made by the authors and fundamental information about the compounds are missing. The manuscript will be of some interest to the readers of Molecules.

The manuscript qualifies for publication in Molecules after major revision.

Line 68-74: there is no information about how the different compounds used in this study were selected. What parameters were used?

Line 81-83: the figure does not add any useful information to the manuscript.

Line 84: the molecular docking data need to be discussed in much more detail- what do the best eight compounds have in common? How do they interact with the ATP-binding site? An illustration would be appropriate.  

Line 102-104: It would be much more informative to see how the compounds dock to the protein structure than a listing of the contact residues.

Line 128: figure 4A shows single concentration inhibition at 1µM and erlotinib gave 57% inhibition. However, the IC50 for erlotinib was determined to 26 nM (figure 4b). It is surprising that the 1 µM concentration only lead to 57% inhibition (38-fold higher concentration than the measured IC50 value).

Line 134-137: the authors report IC50 values as a measure for cytotoxicity-this is not correct (IC50 = half-maximal inhibitory concentration). In section 3.6. Cytotoxicity in cancer cell lines, the use of an MTT assay is described. However, the data from this assay is not shown or discussed in the manuscript.

The manuscript does not include any selectivity date for VF16. Compound selectivity is an important issue when developing new bioactive compounds. In many instances, a lack of selectivity can translate to increased toxicity. Protein kinases are particularly concerned with this issue because they share high sequence and structural similarity. The selectivity of VF16 should be assessed by using a protein kinase-profiling panel.

There are several typographical errors in the manuscript. The authors should find a native English speaker to proofread the manuscript.

Author Response

Dere Reviewer1

I submit the response to your comments. Please see the attachment.

Best regards

Thitinan Aiebchun

Reviewer 2 Report

The manuscript submitted by Aiebchun et al. reports novel vinyl sulfones as inhibitors of wild type EGFR. The authors have used docking to prioritize compounds for experimental testing and investigated selected compounds in an in vitro assay of EGFR kinase activity. The most potent compound VF16 was further evaluated in a cell proliferation assay using two different human cancer cell lines. In addition, the binding mode of this compound was analysed in a 500ns MD simulation and the simulation results were analysed in detail to identify the most important residues responsible for VF16 binding.

While there is some merit in this manuscript, I have a number of concerns that need to be addressed before it is suitable for publication:

1) In the introduction, the authors explain in detail how EGFR overexpression leads to cancer. I would suggest to also include a comment about the relevance of common oncogenic EGFR mutations such as exon 19 deletions or the L855R mutation. This is especially important, as the EGFR inhibitors erlotinib and gefitinib– explicitly mentioned in the manuscript on page 2, line 62 – are only approved for treatment of patients harbouring such EGFR mutations (see here for details: https://www.fda.gov/drugs/resources-information-approved-drugs/erlotinib-tarceva; https://www.accessdata.fda.gov/drugsatfda_docs/nda/2015/206995Orig1s000Approv.pdf). Please also note that to the best of my knowledge, lapatinib is approved for Her2 positive breast cancer, and not cancers with EGFR aberrations (although it potently inhibits EGFR) (https://www.accessdata.fda.gov/drugsatfda_docs/label/2018/022059s023lbl.pdf).

2) On page 2, lines 64-66 the authors write: “In addition, acquired drug resistance caused by the secondary mutation T790M of EGFR tyrosine kinase (TK) domain develops inevitably after a median response duration of 9 to 13 months [18].” Please note that reference 18 (Yun, C.-H., et al., The T790M mutation in EGFR kinase causes drug resistance by increasing the affinity for ATP. Proceedings of the National Academy of Sciences, 2008. 105(6): p. 2070.) does not contain any statements regarding time to drug resistance.

3) In the introduction and the abstract, emphasis is put on mutant (and drug-resistant) EGFR. However, all experiments in the manuscript were conducted using wild type EGFR and the compounds have not been tested on EGFR mutants. The abstract and introduction are therefore misleading.

4) Please also include erlotinib as a reference in Figure 3.

5) What was the rationale behind selecting the particular docking cut-off indicated in Figure 3?

6) On page 4, line 96-97, the authors state: “However, none of the vinyl sulfone derivatives is stronger than erlotinib binding in the ATP-binding site.” Does this statement refer to the binding free energies predicted in the docking studies? Please clarify.

7) Please include a Figure directly comparing the VF16 docking pose (or a refined pose retrieved from the MD simulation) to the erlotinib binding pose.

8) Is the % Inhibition normalized to DMSO control in Figure 4A? If not, please include the DMSO control in Figure 4A. Can you please clarify what “positive” and “negative” means in Equation 1?

9) On page 7, line 137, the authors state: “The VF16 significantly inhibited the A431 lung cancer cell line better than A549 cells.” Please provide statistical details whenever you report “significant” differences.

10) On page 9, line 169, the authors state: “The free energy calculation based on MM-GBSA method was applied to predict the binding affinity of VF16/EGFR-TK complex. We found that the ΔGbind of VF16 is almost identical to the ΔGExp (Table 4).” Please note that no Table 4 exists in the manuscript.

11) On page 10, line 179, the authors state: “… and we found that the -OCH3 moiety of VF16 strongly formed H-bond with C797 and weakly formed H-bond with M793. This is in good agreement with the previous reports showing that H-bond formation with M793 is the main interaction of erlotinib and gefitinib in complex with EGFR-WT and EGFR-MT [34-37].” Are you to referring to the M769 reported in Table 1? Please make sure that the residue numbering is consistent throughout the manuscript.

12) On page 10, line 196-197: “For system validation, the crystalized ligands were defined as a center in the active site for redocking using CDOCKER programs.” Please report the RMSD value obtained in the course of re-docking. Please also describe how the protein has been prepared for docking.

13) Page 6, section 2.3: Please include the assay technology (i.e. the name of the assay). At the moment, it is not clear how compound inhibition has been measured when reading this results section.

In general, the manuscript would benefit from a language editing service and it should be checked for number of spaces.

Author Response

Dere Reviewer2

I submit the response to your comments. Please see the attachment.

Best regards

Thitinan Aiebchun

Round 2

Reviewer 1 Report

The revised version of the manuscript entitled Identification of vinyl sulfone derivatives as EGFR tyrosine kinase inhibitor: In vitro and in silico studies, still need additional modifications before it can be published.

  • The authors have still not provided information about how the different compounds used in this study were selected. They write (lines 72-76): Vinyl sulfones (VFs) are organic compounds, where the core structure of vinyl sulfones is similar to chalcones [22-25] (Figure 1); thus, we hypothesized that vinyl sulfone derivatives can inhibit EGFR-TK activity in a manner similar to chalcones. In this study, we aimed to find new potential anti-cancer agents against EGFR-TK. A series of designed vinyl sulfone derivatives was initially screened by molecular docking technique. How did they select the vinyl sulfone derivatives used for the docking?
  • The authors write (lines 152-157): We found that the inhibitory effect on three cell lines by MTT assay of VF16 (IC50 of 54.63 ± 0.09 µM, 33.52 ± 2.57 µM and 30.38 ± 1.37 µM for A549, A431, and H1975, respectively) is similar to that of the erlotinib (IC50 of 48.21 ± 7.43 µM and 27.19 ± 6.93 µM 155 for A549 and A431, respectively).……I would recommend the authors to have a look at for example https://www.abcam.com/kits/mtt-assay-protocol to see how these kind of data normally is presented.
  • The manuscript does not include any selectivity date for VF16. Docking VF16 into two other kinases is NOT a way to address selectivity. The authors need to show experimental data demonstrating a reasonable selectivity profile.
  • There are still several typographical errors in the manuscript. The authors should find a native English speaker to proofread the manuscript.
